# Stereocilia Rootlets: Actin-Based Structures That Are Essential for Structural Stability of the Hair Bundle

**DOI:** 10.3390/ijms21010324

**Published:** 2020-01-03

**Authors:** Itallia Pacentine, Paroma Chatterjee, Peter G. Barr-Gillespie

**Affiliations:** Oregon Hearing Research Center & Vollum Institute, Oregon Health & Science University, Portland, OR 97239, USA; pacentin@ohsu.edu (I.P.); chatterp@ohsu.edu (P.C.)

**Keywords:** rootlet, actin, stereocilia, hair cell

## Abstract

Sensory hair cells of the inner ear rely on the hair bundle, a cluster of actin-filled stereocilia, to transduce auditory and vestibular stimuli into electrical impulses. Because they are long and thin projections, stereocilia are most prone to damage at the point where they insert into the hair cell’s soma. Moreover, this is the site of stereocilia pivoting, the mechanical movement that induces transduction, which additionally weakens this area mechanically. To bolster this fragile area, hair cells construct a dense core called the rootlet at the base of each stereocilium, which extends down into the actin meshwork of the cuticular plate and firmly anchors the stereocilium. Rootlets are constructed with tightly packed actin filaments that extend from stereocilia actin filaments which are wrapped with TRIOBP; in addition, many other proteins contribute to the rootlet and its associated structures. Rootlets allow stereocilia to sustain innumerable deflections over their lifetimes and exemplify the unique manner in which sensory hair cells exploit actin and its associated proteins to carry out the function of mechanotransduction.

## 1. Introduction

Eukaryotic cells use actin as a basic building block of the cytoskeleton. Actin monomers bind end-to-end to one another and form filaments, which grow through a polymerization reaction that is controlled by ATP [1]. All actin filaments are polarized, with filament ends that are named for their appearance when decorated with a myosin motor fragment; polymerization occurs about ten times faster at the barbed (or plus) end than it does at the pointed (or minus) end. Depending on how filaments are organized, actin can be deployed for a variety of cellular processes. Cells use actin to power cell motility or control cytokinesis [2]; cells also exploit actin to adopt and maintain specific cellular structures [3,4], which include lamellipodia, filopodia, microvilli, and stereocilia. Adding another layer of versatility, compartments of actin can form within larger actin frameworks simply by differential organization of the filaments; these specialized compartments can then perform unique functions. One such compartment is the rootlet.

Actin-containing rootlets anchor actin-based structures to the cell’s soma, and these rootlets can be found associated with microvilli, stereocilia, and other actin protrusions [5,6,7,8,9,10,11,12]. By their appearance, rootlets appear to secure those actin protrusions to the rest of the cell. Confusingly, the term “rootlet” is also used to describe anchors for microtubule-based processes, including cilia and flagella [13,14,15,16,17]. Cilia and flagella emanate apically from the basal body, an organizing structure for microtubules, whereas rootlets project from the basal body deep into the cell [18]. Ciliary rootlets are thought to be formed by rootletin [19,20], which is also a structural component of basal bodies and centrosomes. Microtubule-associated rootlets are not thought to have any molecular similarities to the rootlets of actin-based processes.

Both types of rootlets share several key features, including an internal organization that is distinct from surrounding structures, a role in anchoring projecting structures of cells, and specialized binding proteins. We focus here on the essential role of the rootlet in the stereocilia of sensory hair cells.

## 2. Hair Cells Mediate Hearing and Balance

Hair cells are specialized neuroepithelial cells that are responsible for detecting sound and head movements [21]. Hair cells of the auditory system are functionally similar to those of the vestibular system, capable of responding to deflections of their apical hair bundles by external forces in both cases. Auditory and vestibular stimuli are uniquely conveyed to hair cells, accounting for the differential sensitivity of the two systems to differing external stimuli. Hair bundles each contain 20–300 stereocilia per hair cell [22], arranged in rows of increasing height. Adjacent stereocilia are connected to each other by a variety of linkages, the most famous of which are the tip links [23], which gate the mechanoelectrical transduction (MET) channels [24]. Deflection of the hair bundle leads to sliding of individual stereocilia relative to their neighbors, which in turn increases tension on tip links, activating the channels. MET channels are nonselective cation channels, and by admitting K^+^ from the special ionic environment that is exposed to the apical surface (the endolymph), channel opening leads to hair cell depolarization. In turn, depolarization activates neurotransmitter release at the base of the hair cell, leading to excitation of auditory or vestibular nerves.

## 3. Compartments of Actin in Sensory Hair Cells

Hair cells compartmentalize actin into three main structures: the stereocilia, a cuticular plate, and the rootlets (Figure 1), each of which has a specialized role in the overall function of the hair cell. These compartments are defined by type of actin filament and how these filaments are arranged.

The most visually striking compartment of a hair cell is the stereocilium, a finger-like extension from the apex of the soma. Stereocilia, which are comprised of parallel actin filaments that are heavily crosslinked, are very rigid and do not bend except at the base [25]. All stereocilia actin filaments are oriented in the same direction, with the barbed end pointed away from the cell [26]. The number of actin filaments decreases from hundreds or even thousands along the shaft to only a few dozen at the base of the stereocilium where it inserts into the apical soma [5]. This reduction in filament number shapes the base into a narrow taper. This taper allows the stereocilium to pivot [27], which is the mechanical movement that is required to activate a hair cell and give rise to hearing and balance [21].

The second compartment is the cuticular plate, which is located below the stereocilium in the apical soma [28]. The cuticular plate is a bowl-shaped mesh of actin, constructed of filaments with no directionality [7]; pointed and barbed ends are randomly oriented, and are crosslinked with proteins like spectrin [29], LMO7 [30], and XIRP2 [31,32]. This meshwork of filaments provides a solid structure similar to the way entwined twigs allow a bird nest to hold its shape, and makes an ideal anchoring point for a third actin structure, the rootlet.

## 4. Morphology of the Rootlet

Hair cell rootlets were first noted in 1965 in transmission electron microscopy (TEM) images, where they appear as dark, splinter-shaped densities associated with stereocilia [33]. Rootlets are osmiophilic and therefore appear as heavily stained structures in conventional TEM micrographs (Figure 1). A single rootlet appears as a core within a stereocilium, starting halfway down or lower, and extends down through the tapered region and into the cell soma, anchoring into the cuticular plate (Figure 2a). The rootlet thus forms a bridge that connects the stereocilium and cuticular plate. Rootlets contain uniquely organized actin [34,35,36], with a higher density of filaments than in either the stereocilia or the cuticular plate.

Rootlets begin to develop after the initial stereocilia lengthening phase [37]. In the chick auditory organ, rootlets begin to appear at embryonic day 13 (E13), forming simultaneously with the taper [6]. In some cases, stereocilia actin filaments appear to merge into the rootlet, suggesting that formation of the rootlet may contribute to shaping of the taper [38]. By E14, an osmiophilic plug is visible at the stereocilia insertion; rootlet filaments continue to grow within the cuticular plate for several more days, forming the lower rootlet. In the mouse auditory organ (the cochlea), rootlets begin to develop around E15, and the cuticular plate then forms around the rootlet by E18 [39]. Others have not observed rootlet development in mice until after P1, with the upper rootlet (the dense core at the base of stereocilia) appearing first, and then extending down through the taper into the cuticular plate [40]. These developmental differences may reflect the lack of a strict morphological definition for the rootlet.

Detailed TEM images have revealed the shape and structure of the rootlet. In longitudinal sections, the rootlet is composed of parallel filaments, similar to but more densely packed than in the stereocilium [34,35,41]. When examined by TEM after staining with tannic acid, which can act as a negative stain, the filaments stain relatively lightly but show a periodic banding pattern, occurring at approximately 36 nm intervals [35]. As 36 nm is the actin helix half-repeat, the periodicity at which actin subunits are optimally oriented for interaction with binding proteins [42], these results suggest the presence of a regular structure binding the rootlet together. In alligator lizards, rootlets are demonstrably composed of actin [5,7], but the identity of filaments in mammalian rootlets was initially unclear. While the dimensions and organization of the filaments resembled actin paracrystals [34,35], in early experiments with mammals, rootlets did not label with S1 myosin fragments [25,26,41]. Tight packing of the filaments likely impeded entry by S1 myosin fragments; however, thin sectioning exposed the rootlet filaments, which could then be labeled with anti-actin antibodies [36]. Mammals have six genes for actin [43], two of which—*ACTB* and *ACTG1*—are expressed in non-muscle cells. Immunogold labeling revealed that rootlets are primarily comprised of ACTB oriented with barbed ends up, exactly like the stereocilia, though they label sparsely for ACTG1 too [38,44].

Rootlets vary in their morphology as they traverse the stereocilia and cuticular plate. In alligator lizards, lower rootlets are composed of filament ribbons held in a circular pattern by fibrils that are 3 nm in diameter, forming a hollow tubular structure [5,7]. These fibrils do not label with S1 myosin fragments, suggesting that they are not actin [5,7]. However, in horizontal sections of mammals, the rootlet is composed of approximately 8 nm diameter filaments that are arranged in a hexagonal pattern in both the upper and lower rootlet [34,35]. Some lower rootlets have a hollow center [34,35], particularly those from the first and second row stereocilia [38]. The filaments stain lightly with tannic acid and are outlined by electron-dense material [41]. With a center-to-center spacing of about 8 nm [34,40], close to the diameter of the filaments themselves [35], it is unlikely that actin binding proteins are present between the actin filaments of mammalian rootlets.

In the mammalian inner ear, horizontal TEM sections show that rootlets within the stereocilium (upper rootlets) have a dense core surrounded by a dense ring, with a gap of non-dense material in between [38]. In outer hair cells, the rootlet is widest just above the entry point to the cuticular plate [38]. The rootlet diameter corresponds to stereocilia height, with longer stereocilia having thicker rootlets [38]. As the rootlet passes down through the taper region, the dense ring converges on the rootlet, so that lower rootlets appear as a solitary dense core with a non-dense gap circling it [38]. In the non-dense gap surrounding lower rootlets, thin fibrils (3–5 nm in diameter) extend outward radially from the rootlet, connecting it to the meshwork of the cuticular plate (Figure 2b) [34,35,38,41,45]. These radial fibrils are most numerous at the apex of the cuticular plate, with the number and length of fibrils decreasing as the rootlet plunges deeper [45]. Other filaments form rootlet–rootlet connectors within the cuticular plate [34]. Neither the radial fibrils nor the rootlet–rootlet connectors decorate with S1 myosin fragments, which are specific for actin filaments, suggesting that these filaments are comprised of proteins distinct from actin [41,45].

In the mouse auditory organ, rootlet position and length vary by hair cell subtype, location of the cell along the sensory epithelium, and by individual rows of stereocilia. In all hair cell types, the total length of a rootlet correlates with the height of its associated stereocilium; the longest rootlets in an individual cell are associated with first row stereocilia, the tallest row [38,40]. Loss of rootlets does not affect stereocilia height, nor does loss of stereocilia affect lower rootlet length, indicating that independent mechanisms control the lengths of these two structures [40]. The longest rootlets, those associated with first row stereocilia, sometimes punch through the cuticular plate entirely [38] and have been observed to take a sharp 110° turn to merge with filaments of the striated organelle [46]. The striated organelle is a cytoskeletal lattice located near the plasma membrane below the cuticular plate; it is only found in mammalian inner hair cells and vestibular hair cells [47,48,49]. In outer hair cells, the apical soma folds over to form a lip that contains the outer edges of the cuticular plate, and lateral rootlets within this lip have been observed to angle sideways and directly contact the lateral wall [38]. At the base of rootlets within the cuticular plate, some rootlets become thin, flat, or crescent-shaped, and the terminal points can sometimes splay [38].

Another important measure is how far the upper rootlet extends into the stereocilium; greater penetration into the stereocilium may indicate additional strength. In all apical mouse auditory hair cells, both inner and outer, upper rootlets rise about a third of the way up the stereocilia, whereas in basal outer hair cells, they rise about halfway up the stereocilia [38]. This measurement has not been reported for basal inner hair cells. In addition, how deep the rootlet extends into the cuticular plate may reflect the stresses imposed on the stereocilium. In one study [38], reconstructed serial vertical TEM sections were used to measure the lengths of lower rootlets, and this length was compared to the height of their stereocilia as a ratio (stereocilia height: lower rootlet length). In inner hair cells, regardless of location, there was a consistent drop in this ratio across stereocilia rows. This progressive drop in ratio across the stereocilia rows also occurred in apical outer hair cells, where the first, second and third row ratios were 2.7:1 2:1, and 1.7:1. This shifting ratio may suggest that the length of lower rootlets has an upper limit, possibly set by the size of the cuticular plate. Basal outer hair cells showed a uniquely consistent ratio across stereocilia rows; in all rows, the height of the stereocilium was about equal to the length of the lower rootlet (ratio about 1:1).

It is intriguing that there are differences in the morphology of rootlets when comparing basal and apical hair cells. Mammalian auditory hair cells are frequency-tuned, with apical cells responding best to low frequencies and basal cells responding to high frequencies. In basal outer cells, the longer upper and lower rootlets (relative to stereocilia height) may reflect a greater need to stiffen and strengthen stereocilia in high-frequency cells, which are subjected to many more cycles of stimulation. These differences in rootlet organization among hair cell populations could contribute to frequency selection.

Rootlets can bend in the taper region of the stereocilia [38] where the stereocilia pivot; bending makes functional sense, as activation of the hair cell relies on the stereocilia pivoting. Strikingly, however, rootlets are bent even at rest—the lower rootlet can be angled as much 7° off from its upper section and corresponding stereocilium [50]. Furthermore, the direction of this angle differs across stereocilia rows. The angles of rootlets in tall stereocilia are negative (bent towards shorter stereocilia), and those in short stereocilia are positive (bent towards taller stereocilia). This bending suggests that rootlets contribute to forces that cause the stereocilia to maintain contact with their neighbors by leaning into each other, taller toward shorter and vice-versa. Previously, it was thought that the curve of the cuticular plate into a shallow bowl was the only contributor to this resting tendency for stereocilia tips to contact one another [51]. Further investigation revealed that while rootlets may contribute to the resting lean of stereocilia, they are likely not required for maintaining this contact during deflections. During deflections, the bundle integrity is instead attributed to side and ankle linkages between neighboring stereocilia [50]. Internal tension within the hair bundle caused by motor control over the tip links could lead to bending of rootlets [52], as could tension that pulls rootlets apart from each other, especially if they are connected by elastic linkages. Regardless, the observation that rootlets are bent at rest and presumably impart internal tension to the bundle suggests that forces that keep stereocilia spaced at discrete intervals are also very important.

## 5. Rootlet Protein Composition

While prior studies utilized TEM imaging of the stereocilia rootlets to determine their structure, reports of the protein composition of the rootlet—beyond actin—have only recently emerged (Figure 2c). Two groups of proteins associated with rootlets can be distinguished: those that are internal components, and those that associate externally with the rootlet. Although the filamentous component of the rootlet is actin, several other proteins have been localized to the rootlets directly. An early study examined the localization of contractile proteins in the mammalian auditory hair cells [36]; antibodies against tropomyosin, an actin binding protein, labeled the rootlet region of both outer and inner hair cells. Although tropomyosin’s precise localization relative to the rootlet actin density was unclear, later immunogold labeling experiments confirmed tropomyosin’s presence within stereocilia rootlets [38]. Tropomyosin binds within the groove of the actin helix and prevents binding of some proteins to actin [53]; in the rootlet, tropomyosin may control access of actin binding proteins to the rootlet structural filaments.

A key structural component of rootlets is TRIOBP (TRIO and F-actin binding protein), an actin-associated protein [54]; highlighting its significance for the auditory system, mutations in *TRIOBP* cause hereditary hearing loss in humans [55,56]. The mouse *Triobp* gene encodes three isoforms: TRIOBP-5 is full-length, while TRIOBP-4 only encodes the N-terminus and TRIOBP-1 the C-terminus [57,58]. TRIOBP-1 is ubiquitously expressed, whereas the expression of TRIOBP-4 and TRIOBP-5 is mainly restricted to the eye and inner ear [55,56]. In vitro, TRIOBP-4 tightly bundles actin into rootlet-like structures [40,58]. While loss of TRIOBP-1 causes embryonic lethality, mice expressing TRIOBP-1 but lacking both TRIOBP-4 and TRIOBP-5 have fragile stereocilia that never develop rootlets, and that are prone to damage [40,58]. Loss of just TRIOBP-5 causes dysmorphic rootlets, with lower rootlets that are thin and wispy or absent, and upper rootlets that are elongated and widened [58].

Immunolabeling demonstrated that TRIOBP-4 is present primarily in the upper rootlet, while TRIOBP-5 expression is restricted to the lower rootlet [40,58]. A GFP fusion with TRIOBP-1 localizes to rootlets [59], suggesting that the C-terminus of TRIOBP-5 (which contains the complete TRIOBP-1 sequence) interacts with binding partners while the N-terminus (which contains actin-bundling domains) interacts with the rootlet actin filaments. TRIOBP expression is reduced in the absence of LIM-only protein 7 (LMO7), a primary component of the cuticular plate [30]. Consistent with loss of TRIOBP-5, LMO7-deficient mice also have abnormal rootlet morphology, and suffer progressive hearing loss [30].

Pejvakin (PJVK) also localizes to rootlets. While mice lacking PJVK suffer from profound hearing loss, their rootlets appear normal, and instead hair bundle morphology is disrupted [59]. TRIOBP co-localizes with PJVK, and TRIOBP-1 interacts with the C-terminus of PJVK [59]. As the exons encoding TRIOBP-1 are also found in TRIOBP-5, it is likely that PJVK is a binding partner of TRIOBP-5. The functional role of PJVK in rootlets is unclear, however.

Taperin (TPRN) may also play a role in rootlet function. TPRN is primarily concentrated at the taper region of the stereocilia and associated with non-syndromic hearing loss in humans [60]. Loss of TPRN in mice leads to the disruption of the stereociliary rootlet and eventual loss of stereocilia, which results in hearing loss [61]. Visualization of TPRN using stochastic optical reconstruction microscopy (STORM) suggested that TPRN is present in the core of the taper, where the rootlet resides [62]. Although TPRN immunoreactivity does not appear to be associated with the lower rootlet within the cuticular plate, *Tprn^−/−^* mice have rootlets that are unusually curved and that have hollow central regions surrounded by dense rings on the periphery [61]. TPRN can have profound effects on stereocilia actin structure; when the *Grxcr2* gene is disrupted, TPRN mislocalizes to upper stereocilia shafts, and stereocilia are both longer and profoundly disrupted [63]. The tight restriction of TPRN to stereocilia bases is thus functionally important.

Experiments using immunogold labeling suggest that rootlets contain the actin-binding protein spectrin [38]. Spectrins usually oligomerize as αβ tetramers, and the predominant isoforms expressed in mouse hair cells are SPTAN1 and SPTBN1 [64]. A recent study imaged both SPTAN1 and SPTBN1 using super-resolution fluorescence microscopy [65], and showed that spectrin is not a structural component of rootlets, but rather surrounds the lower rootlet. While spectrin is initially present throughout the cuticular plate in both inner and outer hair cells, by P14, spectrin condenses into ring-like structures that surround lower rootlets, extending several hundred nanometers down into the cuticular plate [65]. The spectrin structures are hollow cylinders or sheaths that surround the rootlets. The late formation of the spectrin sheaths suggests that this protein has a role in maintenance of rootlets. Spectrin sheaths are not found in the third row of inner hair cells [65]; since these stereocilia pivot when the hair bundle is deflected, spectrin cannot be responsible for the ability of the dense rootlet to bend in the taper region. Instead, it suggests a role in strengthening the connection between the lower rootlet and the cuticular plate. The stereocilia of the third row are narrow compared with those of the first and second rows, so the extra support provided by spectrin may only be necessary for thicker stereocilia. This conclusion is supported by spectrin distribution in vestibular hair cells. In these hair cells with thinner stereocilia, spectrin forms a meshwork in the cuticular plate instead of ensheathing the lower rootlets [65].

Mice lacking SPTBN1 display severe deafness, which highlights the importance of spectrin sheaths for mammalian hearing [65]. SPTBN1 is also required for proper localization of SPTAN1, and so SPTAN1 is also missing from cuticular plates in *Sptbn1^−/−^* mice [65]. TPRN expression is disrupted by the loss of SPTBN1 [65], although the altered distribution reflects the disorganization of stereocilia in *Sptbn1^−/−^* mice rather than a change in TPRN distribution within a stereocilium. Spectrin also plays an important role in the localization of RIPOR2, also known as FAM65B, around the rootlet [65]. RIPOR2 is expressed in a distinct ring around the edges of the taper region, with no labeling in the center [62]; in *Sptbn1^−/−^* mice, RIPOR2 is no longer associated with the stereocilia insertions. RIPOR2, like spectrin, apparently surrounds rootlets, and its altered distribution in *Sptbn1^−/−^* mice suggests that spectrin anchors it there. While it is unclear whether RIPOR2 has a distinct function in rootlets, its presence influences the localization of TPRN; in *Ripor2* knockouts, TPRN no longer localizes to rootlets [62]. Similarly, TPRN localization to the base of stereocilia is dependent on the protein CLIC5. TPRN and CLIC5 form a complex with RDX and MYO6 [66], and the three proteins other than TPRN are concentrated in the taper region but are not thought to associate directly with rootlets.

In the frog saccule, immunoelectron microscopy revealed that MYO1C, an unconventional myosin, was located in a ring where stereocilia insert into the soma [67]. Additionally, these authors also showed that isolated stereocilia display a concentration of the unconventional myosin MYO6 at their tapered ends [67]. Combined with its known association with TPRN, MYO6 may be involved in rootlet formation or function. Whether myosin molecules can crawl along exposed rootlet actin filaments is unclear; both MYO1C and MYO6 may be bound to rootlet actin filaments but are unable to move further along the filaments, trapping them at the base of stereocilia.

In the auditory organ of guinea pigs, TEM with immunogold labeling demonstrated that calmodulin (CALM) is enriched in the rootlet region of both outer and inner hair cells [68]. These results suggest that a component of the rootlet may bind CALM and be regulated by Ca^2+^, although no known CALM binding proteins have yet been localized to rootlets.

## 6. Function of the Rootlet

Projecting structures often have rootlets or similar structures, and it is likely that the fundamental function of a rootlet is to anchor the projection. A dense core that literally roots into the principal cytoskeleton may provide stability and strength to otherwise flimsy structures. Hair cell stereocilia undoubtedly rely on rootlets to resist damage, as stereocilia that lack rootlets are more susceptible to long-term damage after deflections [40]. Rootlets could resist damage either by strengthening the stereocilia insertion or by increasing stereocilia stiffness, preventing large damaging deflections. Stereocilia that lack rootlets are 2–4× more flexible during deflections by fluid-jet stimulation; however, once BAPTA treatment removed the contribution of extracellular linkages between stereocilia, stereocilia without rootlets were then 3–10× more flexible than controls [40]. Since neither TEM nor scanning electron microscopy revealed extra bending in stereocilia lacking rootlets, the change in flexibility must arise from alterations to the mechanical properties of the taper region rather than to changes in the stereocilia core rigidity [40]. Even with its specialized reinforcing structures, or perhaps because it is the first line of defense to mechanical stress, the rootlet is the hair cell structure that is most often damaged by sound stimuli that cause permanent threshold shifts [69]. Damaging noise exposure can cause upper rootlets to shorten, which may reduce bundle stiffness, or even to disconnect entirely from the lower rootlet [69]. If rootlets are prevented from developing, stereocilia form and reach normal heights but eventually fuse to their neighbors and degenerate [40]. These observations argue for an essential structural role for rootlets.

Do rootlets contribute to the features of hair cell MET? Hair cells are able to sense sound and vibrations because stereocilia pivot at their bases and slide with respect to each other within the hair bundle; accordingly, detection of motion by stereocilia is the basis of hearing and balance. Experiments and models show that the stiffness of the rootlets (also known as the pivot springs) contributes to overall stiffness of the stereocilia [52,70,71]. The rootlets provide an opposing force that allows the gating spring, the elastic element that controls opening of MET channels, to remain extended at rest. The magnitude of this opposing force can be measured directly in experiments where gating springs are severed with BAPTA iontophoresis while a hair bundle is under displacement clamp [72,73]. Significant rootlet stiffness is also required for the negative bundle stiffness observed in the frog saccule [74,75]. Rootlets thus play a critical role in the mechanics of stereocilia deflection, but are not required to perform MET itself. When applying a displacement stimulus to hair cells using a glass probe, the maximum current elicited from the MET channel was unchanged in *Triobp* mutants, which lack rootlets. Moreover, MET in these mutants did not differ from that of their wild-type siblings with regard to the current–displacement relationship, fast adaptation rate and extent, and slow adaptation rate and extent [40]. Application of a force stimulus to a bundle, for example using a fluid-jet stimulus, reveals the essential function of rootlets; in *Triobp* mutant mice, bundle stiffness was greatly diminished [40]. More in-depth characterization of mechanotransduction using a flexible-fiber force stimulus should reveal the consequences of alteration of the rootlets for hair bundle function.

## 7. Conclusions and Perspectives

In sensory hair cells, rootlets play an essential role in ensuring that the pivot point of stereocilia, the locus of greatest mechanical stress, is durable. Rootlet actin filaments, tightly bundled by TRIOBP and surrounded by hollow spectrin cylinders, must resist significant life-long forces.

Many aspects of rootlets remain mysterious. TEM imaging reveals a rich abundance of filaments that both comprise the rootlet and connect it to other structures, but the identities of the proteins making up these filaments remain largely unknown. For example, which protein comprises the rootlet–rootlet connectors in the lower rootlet, or the radial fibrils? The localization of spectrin makes it a plausible candidate, and in some published images rootlet–rootlet connectors or radial fibrils appear to be labeled by spectrin antibodies [65]. High-resolution immunogold labeling of SPTAN1 and SPTBN1 could determine if the filaments observed in TEM images are indeed spectrin. We still do not know the protein component of the dark ring surrounding the upper rootlet, and there are no good candidates.

The unique mechanical needs for the stereocilia pivot point of the stereocilium stimulate questions regarding the molecular structure of the rootlet. Stereocilia can pivot to extreme degrees, and the rootlet pivots with it. How is it possible for the rootlet to form sharp angles when it is comprised of rigid actin filaments? Rootlets damaged by loud sound often fracture at the pivot point, confirming that this is a weak area [69,76]. Do the actin filaments of the upper and lower rootlets terminate at the pivot point? If so, are they held in place by unknown proteins that provide flexibility? Or is the actin specially organized at this single point so as to allow filaments to bend without breaking? If the bundled rootlet actin filaments can slide with respect to each other (shearing), then differential movements of the filaments can accommodate pivoting stereocilia (Figure 3). Note however that this model, where only rootlet filaments slide but stereocilia filaments do not, contradicts Tilney’s hypothesis that all stereocilia actin filaments shear during a stimulus [77].

The unique cytoskeletal structure of the stereocilia rootlet requires further investigation, both to determine its structural features and to identify the proteins that are required for its assembly, structure, and function. Because of the difficulty in visualizing the three-dimensional structure of the rootlet using conventional TEM methods, the focused ion beam-scanning electron microscopy (FIB-SEM) approach taken by Katsuno and colleagues [58] should be a superior method for rootlet characterization [78], as is electron cryo-tomography [79]. Both methods allow for three-dimensional reconstruction of structures in a volume once segmentation has been carried out. Localization of proteins using antibodies is possible with each technique, but is difficult.

Like other actin domains in hair cells, the rootlet is a rich source of intriguing cytoskeletal mysteries. Further investigation into the rootlet’s structure is essential and now possible given the superior three-dimensional imaging tools available. Moreover, continued identification of new rootlet components, as well as more accurate localization of all components, is necessary to determine the rootlet’s molecular structure and mechanism of assembly. Finally, a deeper understanding of the rootlet’s function is needed; does it serve to strengthen the stereocilia insertion or simply to stiffen the pivot point? Answering these and related questions will contribute to our deepening understanding of the stereocilia rootlet.

## Figures and Tables

**Figure 1 ijms-21-00324-f001:**
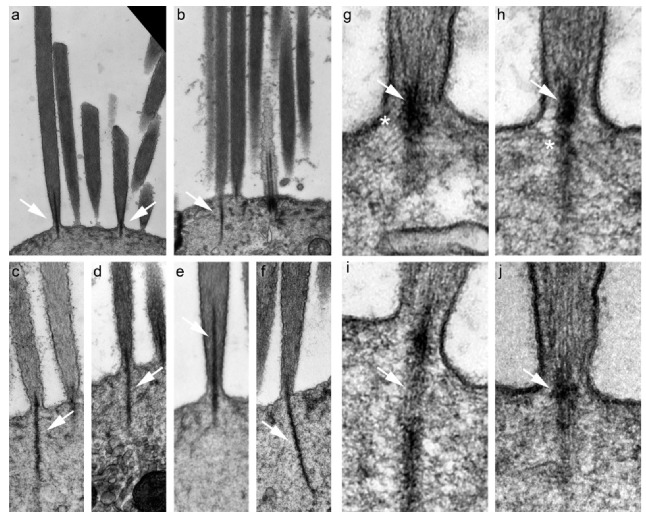
Rootlet structure as visualized with transmission electron microscopy (TEM). Each panel shows an image of a rootlet in a mouse utricle hair cell. Arrows indicate rootlets in each panel; arrows point to upper rootlets in (**a**,**e**), and to lower rootlets in (**b**–**d**,**f**,**i**). Arrows point to the rootlet at the insertion of the stereocilium into the soma in (**g**,**h**,**j**). The apparent length and location of a rootlet depends on the orientation of the thin section used for imaging, as well as the orientation of the rootlet within the stereocilia and cuticular plate. Asterisks in **e** and **f** indicate filaments that appear to connect the rootlet to the membrane and cuticular plate. Ages: (**a**,**b**) P12; (**c**) P5; (**d**–**f**) P12; (**g**–**j**) P5. Panel full widths: (**a**,**b**) 3000 nm; (**c**–**f**) 700 nm; (**g**–**j**) 350 nm.

**Figure 2 ijms-21-00324-f002:**
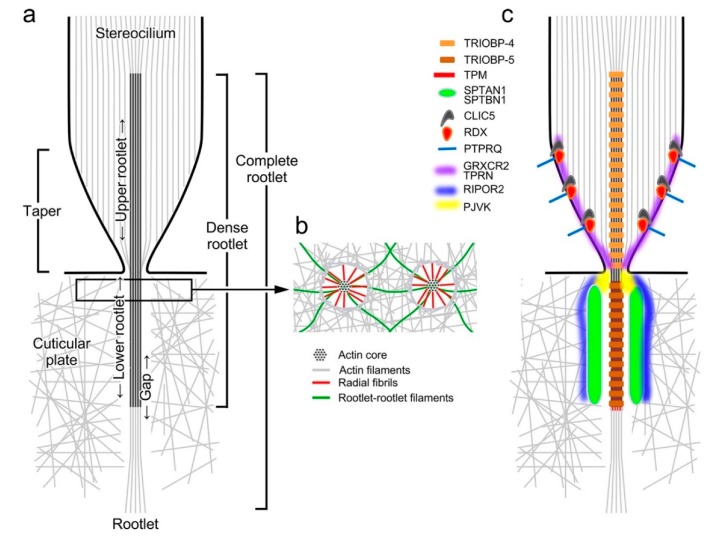
Diagrams illustrating rootlet structure. (**a**) Actin filament structure only. In a stereocilium, spacing of actin filaments (gray) is maintained by actin–actin crosslinkers (not shown). Crosslinkers disappear in the taper region, where most filaments terminate on or near the plasma membrane in a systematic way, forming the taper. By contrast, the central dozen or two filaments are gathered together to form the rootlet, which eventually penetrates into the cell and extends into the cuticular plate. The rootlet stains darkly with osmium tetraoxide, forming the dense rootlet, which accounts for the majority of the rootlet visible in TEM. Rootlets have an upper portion in the stereocilium (‘upper rootlet’) and a lower portion in the cuticular plate (‘lower rootlet’). In the cuticular plate, there is a gap between the filaments of the rootlet and the meshwork of the cuticular plate. (**b**) Rootlet connecting filaments. In cross-sections of the rootlet near the apical surface (see box in **a**), several types of filaments can be identified. Radial fibrils (red) extend from the core of the rootlet to the surrounding cuticular plate, while rootlet–rootlet filaments, visible only after detergent and EDTA extraction prior to fixation, appear to interconnect rootlets. (**c**) Proteins of the rootlet. TRIOBP-4 is exclusively associated with the upper rootlet, while TRIOBP-5 is associated with the lower rootlet. Both TRIOBP splice forms have multiple actin binding domains, which apparently allow TRIOBP to wrap around and bundle the rootlet filaments. TRIOBP-5 has additional domains (separately encoded by the TRIOBP-1 splice form) that may connect TRIOBP-5 to surrounding structures. The spectrin isoforms SPTAN1 and SPTBN1 apparently form a sheath around the lower rootlet, appearing as rings in confocal horizontal sections. CLIC5, RDX, and PTPRQ form a membrane complex in the taper region; they may also bind TPRN, which is also associated with the taper and rootlet. GRXCR2 maintains TPRN in the taper region. The precise localization of PJVK is not clear. RIPOR2 (FAM65B) also forms rings around the rootlet, but it is unclear whether the rings persist along the lower portion of the rootlet.

**Figure 3 ijms-21-00324-f003:**
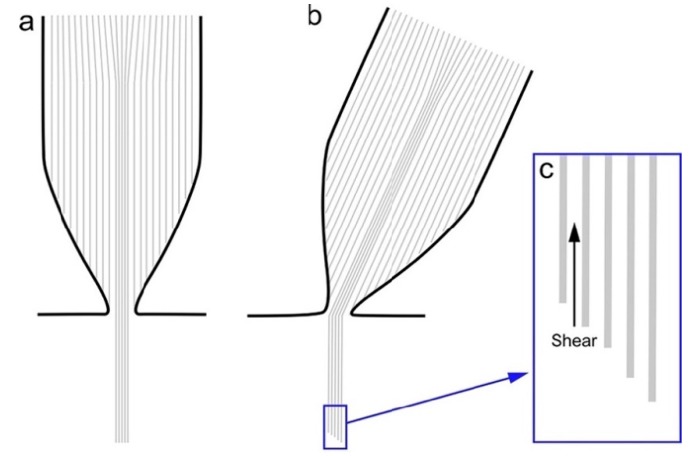
Hypothesis for rootlet filament movement during stereocilia pivoting. (**a**) Rootlet actin filaments are mechanically similar to 1/4” steel cables—flexible but inextensible. Multiple rootlet filaments are bundled together, probably with TRIOBP. (**b**) If rootlet filaments are capable of shearing (sliding with respect to each other), then during stereocilia pivoting, filaments furthest from the direction of the pivoting (left in this diagram) will shear more than those closest to the direction of pivoting. (**c**) Magnified view of the soma ends of rootlets illustrating differential shear of rootlet filaments.

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
