# Peer review of "Stereocilia Rootlets: Actin-Based Structures That Are Essential for Structural Stability of the Hair Bundle"

_ijms, 2020, doi:10.3390/ijms21010324_

Round 1

Reviewer 1 Report

In this review by Pacentine et al, they provide a nice summary of the knowledge surrounding the stereocilia rootlet structure, molecular composition, and function. This is the first review that I know of about the stereocilia rootlet which has been gaining in importance and relevance for hair cell stereocilium function recently and is thus a timely review. I think that the rootlet has important functions and the authors provide some unanswered questions regarding rootlet structure, molecular composition, and function. I have just a few suggestions for the authors which I detail below.

Line 93: “chick cochlea” is not technically correct since the cochlea is in reference to a snail shaped organ which only mammals have. I would use “chick basilar papilla” or “chick auditory organ.”

Line 95-96: “stereocilia actin filaments appear to merge into the rootlet” This is what was observed by Furness et al, however, the diagram in Figure 2a (and all other stereocilia schematics) shows the merging of actin filaments at the top of the rootlet, not in the taper region as shown in the reference, where the stereocilia actin filaments intersect with the rootlet core much lower down. Do the authors have a reason for drawing their diagram with stereocilia actin filaments converging together to create the rootlet at the top of the rootlet? Furness et al suggested the top of the rootlet was pointy. I understand it is just a cartoon but making it as accurate as possible will help the reader.

Line 237: tropomyosin is mentioned to be a structural component of the rootlet, but it is not included in Figure 2c diagram

Line 320-321: I would be careful about the interpretation that “Stereocilia that lack rootlets are more susceptible to long-term damage after deflections.” My understanding is that there is long term damage in the cochlea, but I don’t know if this is due to the lack of the rootlet resisting damage or the rootlet providing stiffness. The stereocilia in the cochlea may be subject to larger deflections when the rootlet is missing due to the decreased stiffness, so the larger stimulations could be causing more damage. It could be that the rootlet provides the stiffness, which then limits the deflection of the stereocilia, which helps to prevent damage. So resisting damage and providing stiffness would be inter-related. I don’t think the experiment where the same size deflections in each genotype is delivered to see whether damage occurs in one case but not another.

If the authors agree, I think some mention of the FIB-SEM technology used by Katsuno et al should be mentioned in terms of studying rootlet ultrastructure. To me, this technology will facilitate greater structural insights by the ability to reconstruct all the rootlets in various mutant mice more easily. FIB-SEM with immunogold labelling is also developed.

Author Response

See file

Reviewer 2 Report

The authors described the review article entitled that Stereocilia rootlets, actin-based structures that are essential for structural stability of the hair bundle. This is an interesting study in an area that needs investigating. This is also a carefully written review, and the findings are of considerable interest.
A few minor revisions are listed below.

Minor point:
1. Line 370 In Figure 3, Line 381, and others: What is “shear”? Is that fluid shear stress applied to stereocilia rootlets? For the reader’s convenience, the authors should describe more detail about the “shear.”

2. There are no scale bars in Figure 1. If the magnification of TEM images is the same, the authors include the scale bar in TEM image.

3. Line 362 The authors should include the reference for “High-resolution immunogold labeling of SPTAN1 and SPTBN1 could determine if the filaments observed in TEM images are indeed spectrin.”, if it was available.

Author Response

See file
